# Candidate silencer elements for the human and mouse genomes

Naresh Doni Jayavelu [1], Ajay Jajodia[1], Arpit Mishra[1] & R. David Hawkins [1✉]

The study of gene regulation is dominated by a focus on the control of gene activation or increase in the level of expression. Just as critical is the process of gene repression or silencing. Chromatin signatures have identified enhancers, however, genome-wide identification of silencers by computational or experimental approaches are lacking. Here, we first define uncharacterized *cis*-regulatory elements likely containing silencers and find that 41.5% of ~7500 tested elements show silencer activity using massively parallel reporter assay (MPRA). We trained a support vector machine classifier based on MPRA data to predict candidate silencers in over 100 human and mouse cell or tissue types. The predicted candidate silencers exhibit characteristics expected of silencers. Leveraging promoter-capture HiC data, we find that over 50% of silencers are interacting with gene promoters having very low to no expression. Our results suggest a general strategy for genome-wide identification and characterization of silencer elements.

[1] Division of Medical Genetics, Department of Medicine, Department of Genome Sciences, Institute for Stem Cell and Regenerative Medicine, University of Washington School of Medicine, Seattle, WA, USA. ✉email: rdhawk@uw.edu

It has long stood that repressive transcription factors can bind to the promoters of genes through silencer elements to inactivate gene expression[1–4]. In 1985, the yeast mating-type loci revealed that distal silencer elements could control gene expression from afar[5]. The role for distal silencer elements in mammals was demonstrated shortly thereafter through a silencer element located several kilobases upstream of the rat insulin gene[6]. It would be a decade later before key experiments identifying a silencer in the intron of the mouse and human *CD4* genes would revealed the important role that silencers can play in lineage specificity and cell fate determination, as this silencer represses *CD4* expression in CD8[+] T cells[7,8]. Later several studies identified genomic sequences with silencer properties, which are opposite of enhancers across many species[9–15]. While dozens of mammalian silencers have been identified, these elements are largely understudied, possibly due to our biased focus on gene upregulation and the poor understanding of those elements with non-promoter locations.

Classic studies of promoter silencers showed that these elements reside next to activating sequences[4]. Just as promoter elements can switch states from activating to repressive due to the presence of silencer elements and the factors bound, many distal *cis*-acting regulatory elements can have dual activity, depending on the set of conditions and cell type. Several previous studies showed this dual activity of enhancers switching to gene repression, or silencing, in another cell type and vice versa[16–19]. For example, GATA1 binding to a *cis*-acting element upstream of the *Gata2* promoter displaces the activating GATA2-bound factor and represses *Gata2* expression[19]. GATA switches at distal elements such as this are common during hematopoiesis[18], and are context, cofactor, and concentration dependent[20–22]. Furthermore, the repressive activity of GATA1 can induce changes in chromatin looping. For example, during hematopoietic differentiation, an upstream enhancer is bound by GATA2 to activate expression of the *Kit* gene in multipotent cells. Subsequently during lineage commitment, GATA1 binds to inactivate the enhancer and also binds a downstream silencer to repress *Kit* expression, which results in loss of the enhancer loop and gain of a silencer loop with the promoter[23].

Outside of promoter regions, silencers along with enhancers and insulators create a complex array of distal *cis*-regulatory elements (CREs). These elements, and the factors that bind to them, are important for the nuanced output of RNA levels across cell types. Identification of distal CREs and our understanding of the regulation of gene expression in mammalian genomes has been greatly facilitated by the genome-wide mapping of element-specific histone modifications or transcription factors, e.g., histone H3 lysine 4 monomethylation (H3K4me1) distinguishing enhancers[24] and CTCF binding to a subset of candidate insulators[25]. While the presence of repressor sequences within CREs such as promoters has recently become more evident by tiling millions of oligos across these regions and testing by massively parallel reporter assays (MPRA)[26], distinct annotations for distal silencer elements in the human and mouse genomes are still missing from our *cis*-regulatory lexicon.

Our goal is to identify distinct silencer elements distal to the genes they regulate. In order to identify silencer elements, first we devised a simple subtractive analysis approach based on DNase hypersensitive sites (DHS) and other known CREs to find the DHS elements lacking known chromatin marks belonging to promoters, enhancers, and CTCF-bound insulators, and we term these elements as uncharacterized CREs. We tested ~7500 of these uncharacterized CREs for silencer activity in K562 cells using MPRA to identify silencer elements. Using MPRA data, we trained a support vector machine (SVM) classifier to predict potential candidate silencer elements from untested

uncharacterized CREs in 82 human and 22 mouse cell or tissue types. This results in a catalog of >1.7 million candidate silencer elements in the human genome and a second catalog of ~1 million candidate silencer elements in the mouse genome. We find that candidate silencer elements are enriched for motifs of known repressive transcription factors, and de novo motifs for potential cognate transcriptional repressors. We are able to validate our predictions by direct silencer interactions with repressed target genes, and functional testing via reporter assays and CRISPR genome editing. Candidate silencer elements are often enriched for disease-associated variants in expected cell types or lineages. These catalogs, which will require more intensive study and validation from the field, should aid our understanding of gene expression through negative regulation of expression.

## Results

### Prediction of genome-wide uncharacterized CREs containing silencer elements.
In order to find genome-wide candidate silencer elements, first we devised an efficient simple subtractive analysis (SSA) approach to determine DHS elements lacking known chromatin marks belonging to promoters and enhancers, or CTCF binding for insulators, and we term these elements as uncharacterized CREs (Fig. 1a). Other potential *cis*-regulatory elements should be present within these regions of open chromatin. We can generate open-chromatin data either from DNase-seq (DNaseI hypersensitive sites sequencing) or from ATAC-seq (assay for transposase-accessible chromatin using sequencing) and other CREs data from ChIP-seq (chromatin immunoprecipitation coupled with sequencing) for any cell type or organism. In this SSA approach, we subtract enhancers (H3K4me1 peaks), promoters (2.5 kb window around TSS and H3K4me3 peaks), and potential insulators (CTCF sites) from open chromatin (DHS) in a cell-type specific manner, and assign the remaining DHS as uncharacterized CREs. We hypothesize that a subset of these uncharacterized CREs will contain silencer elements. Using this SSA approach, we determined 2,315,105 uncharacterized CREs in the human genome spanning across 82 cell types from the Roadmap[27] and ENCODE consortia[28], and 1,299,866 elements in the mouse genome for 22 cell types from the ENCODE consortium[28] (Fig. 1b, c; Supplementary Fig. 1a, b and Supplementary Data 1).

### Uncharacterized CREs enriched with repressor TF motifs and repressor TFBS.
Transcription factors (TFs) bind at CREs and regulate gene expression in response to external cues. Activator TFs bind at enhancers to enhance expression, and repressor TFs bind at silencer elements and repress gene expression. We hypothesized that uncharacterized CREs will contain silencer elements. To validate this hypothesis, we performed TF motif enrichment analysis on uncharacterized CREs across cell types and found a well-known transcriptional repressor, REST is consistently enriched across all tested cell types (Fig. 1d, e; Supplementary Fig. 1c, d). Motifs of USF1, BATF, BACH2, FRA1, ATF3, FOSL2, JUN, NRF2, NFE2, and RFX family of TFs are also enriched and previous studies reported that these TFs display repressor activity[29–32]. In addition to motif analysis, we also checked whether these uncharacterized CREs are enriched for experimentally bound repressor TF-binding sites (TFBS). We intersected uncharacterized CREs with known repressor TFBS within the same cell type using ChIP-seq data for REST, YY1, ZBTB33, SUZ12, and EZH2. We found that 73%, 61%, 49%, and 58% of total uncharacterized CREs in K562, H1, GM12878, and HEPG2 cells, respectively, are typically enriched for one of the above mentioned known repressor TFBS[33–40] (Fig. 1f; Supplementary Fig. 1e). This enrichment of repressor TFBS at

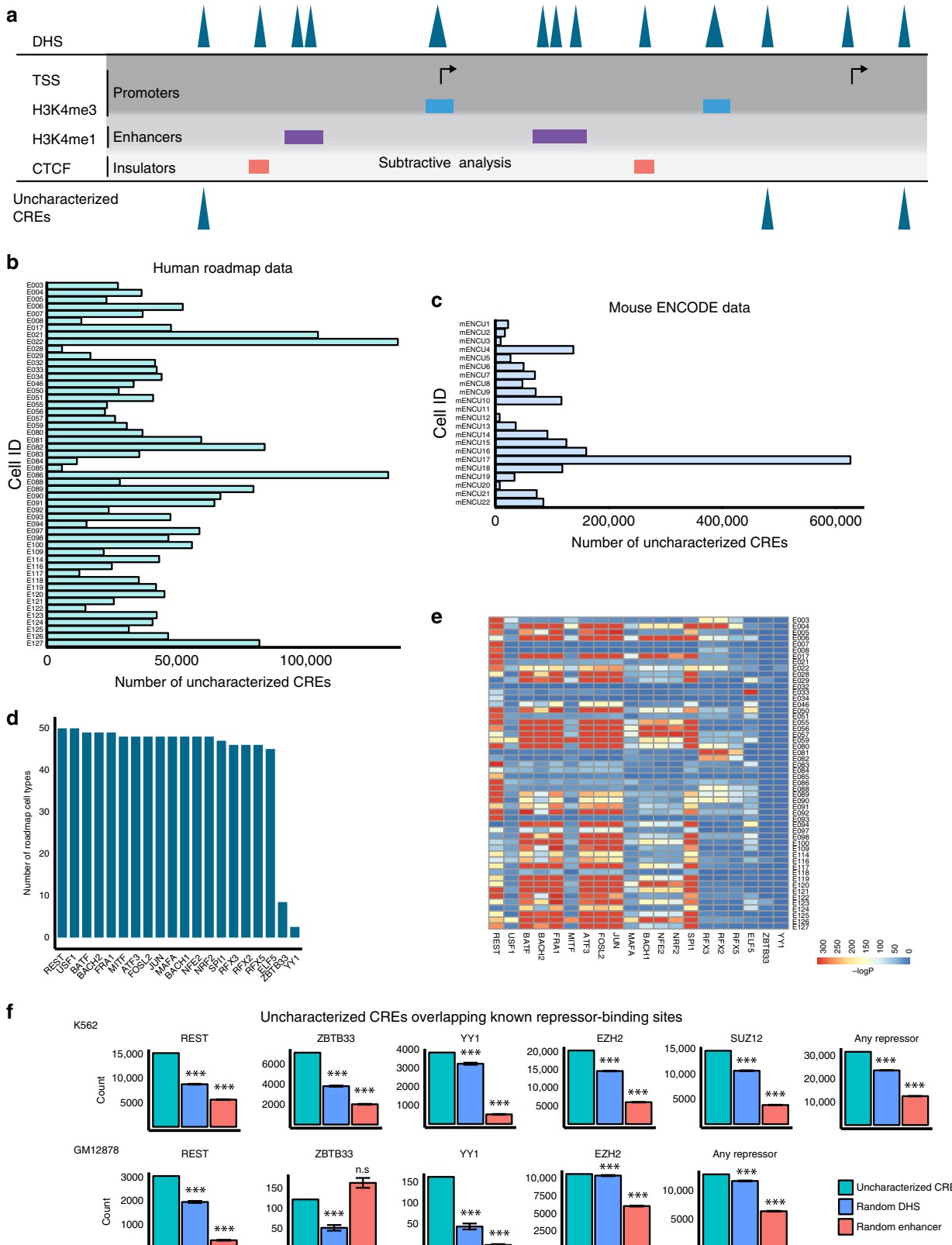

**Fig. 1 Overview of uncharacterized CREs in human and mouse. a** Schematic of the simple subtractive analysis (SSA) approach. Barplots presenting the count of uncharacterized CREs across different cell types and tissues in human from Roadmap (**b**) and mouse genomes from ENCODE (**c**). **d** Bar plot presenting the count of cell types enriched with TF motifs across 52 cell types and tissues from Roadmap. **e** Heatmap of enrichment (-log(P-value)) of TF motifs across 52 cell types. **f** Barplots presenting the count of uncharacterized CREs, random DHS and random enhancers at known repressor TFBS based on ChIP-seq data in K562 and GM12878 cell types. *** indicates permutation test $P$-value < 0.0001, and n.s. denotes not significant.

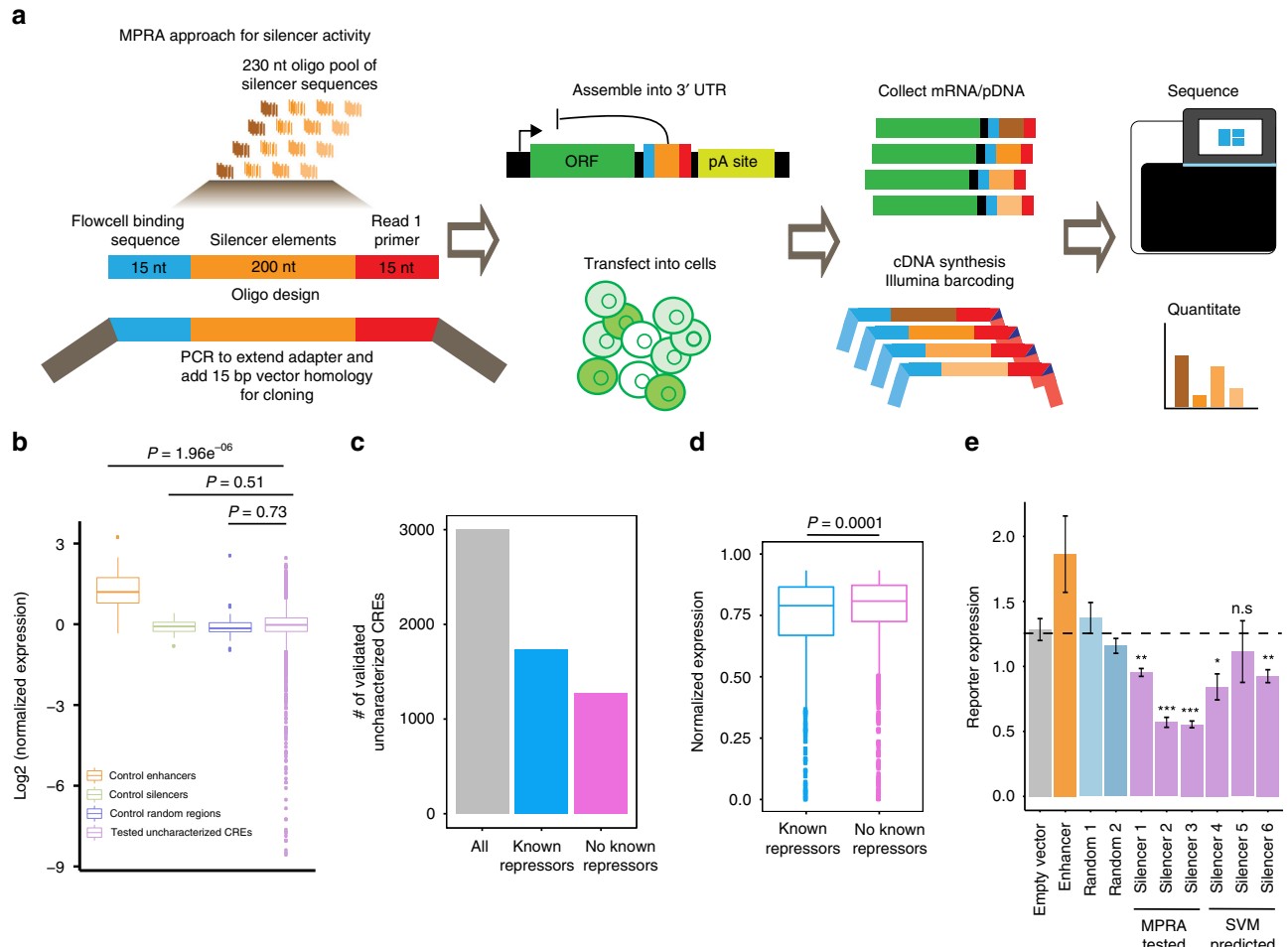

**Fig. 2 Functional testing of activity of uncharacterized CREs in K562 cells. a** Schematic of MPRA/STARR-seq approach for measuring the activity of uncharacterized CREs. **b** Box plot displaying the activity level distributions of enhancers, control silencers, control-random regions, and tested uncharacterized CREs. *P*-values computed by *t* test. **c** Bar plot showing the count of all uncharacterized CREs showing silencer activity, and counts of validated uncharacterized CREs overlapping with known repressor TFBS (TFBS belonging to REST, YY1, ZBTB33, SUZ12, and EZH2) or with other TFs motifs. **d** Box plot comparing the activity level distributions of validated uncharacterized CREs categorized to known repressor TFBS status and other TFs motifs (no known repressor TFBS status). *P*-value computed by *t* test. **e** Bar plot displaying the reporter expression for six silencer elements, two random controls, one enhancer and empty vector (SCP1 core promoter). The dotted horizontal line indicates the mean reporter expression of random regions and acts as a reference. *** denotes significantly lower reporter gene activity of tested silencer than that of corresponding random controls, and n.s. denotes not significant (**P*-value < 0.05, ***P*-value < 0.01, ****P*-value < 0.001, and *P*-values are computed using *t* test). Error bars represent the mean ± s.e.m. of biological replicates. In the box plots, bounds of the box spans from 25 to 75% percentile, center line represents median, and whiskers visualize 5 and 95% of the data points.

uncharacterized CREs is statistically significant compared with random active enhancers and random accessible regions (DHS). This enrichment corroborates that these uncharacterized CREs are distinct from other categories of CREs, and likely contain silencer elements.

**Screening of uncharacterized CREs for silencer elements**. Having shown that the uncharacterized CREs are enriched with repressor TF motifs and repressor TF binding indicative of silencer activity, we proceeded to functionally test a subset of uncharacterized CREs to quantitate their activity for silencer function via massively parallel reporter assays (MPRA) using the STARR-seq[41] approach in K562 cells (Fig. 2a). The human STARR-seq vector utilizes the super core promoter (SCP1), which was designed and shown to be stronger than the CMV promoter[42], therefore, the GFP reporter expression should be susceptible to detectable decreases in expression. We selected 7430

uncharacterized CREs, 20 known silencer elements[26,43], 20 known enhancer elements[26], and 67 randomly selected regions, which act as a control set (Supplementary Data 2). Oligos were designed to span 200 nt of the uncharacterized CREs sequences flanked by 15 nt adapter sequences and synthesized en masse. Of 7430 selected uncharacterized CREs, 3705 elements have at least one TFBS belonging to well-known repressor TFs (REST, YY1, ZBTB33, EZH2, and SUZ12) and 3725 elements have at least one known motif belonging to GATA1, GATA2, GATA3, GATA4, BACH1, BACH2, TCF12, SMAD3, FLI1, RUNX1, KLF4, ZFP187, ZNF263, ZBTB7B, and GFI1B (actual numbers are provided in Supplementary Data 2). All of these TFs were reported to have repressor activity, while primarily serving as activators[29,44–50]. As expected, enhancers showed the highest activity, while control silencers and tested uncharacterized CREs, on average, showed similar levels of activity normalized to the activity of control-random regions (Fig. 2b; Supplementary Fig. 2a). We found 3001 of the tested uncharacterized CREs (1731 elements with known

repressor TFs status, and 1270 elements with other TFs status) to have activity less than the mean activity level of control-random regions at 5% FDR. We call these uncharacterized CREs silencer elements (Fig. 2c). Of uncharacterized CREs demonstrating silencer activity, those elements with known repressor status showed significantly lower activity than elements with other TFs status (Fig. 2d). We also looked at the distribution of TF status in order to identify strong candidate TFBS for potential silencer activity (Supplementary Data 2 and Supplementary Fig. 2b). Though many of TFBS showed roughly 40% success rate, TFBS for REST, EZH2, SUZ12, SMAD3, RUNX1, and GATA family of TFs would be strong candidates indicative of potential silencers. Because most known TFBS motifs are obtained from transcriptional activators, we also performed a de novo motif analysis. The de novo analysis identified several novel TFs motifs. When considering the closest related known motif, these belong to REST, YY1, ZBTB33, and ZBTB3 (Supplementary Fig. 2c).

To validate the silencer activity from MPRA experiments, we tested three silencer elements via traditional reporter assays by cloning the silencer elements upstream of SCP1 core promoter + luciferase gene construct in K562 cells. We noticed a significant decrease in reporter gene expression for three out of three tested silencer elements compared with two random controls (Fig. 2e; Supplementary Data 3).

To further validate our hypothesis that uncharacterized CREs contain silencer elements, we performed CRISPR-Cas9 genome editing in K562 cells. We focused on uncharacterized CREs that are common to GM12878 and K562 cell lines with a known target gene from promoter-capture HiC[51]. We targeted sgRNAs to these uncharacterized CREs regions, hypothesizing that deletion of these elements would result in increased gene expression of the target gene if acting as a silencer element. We observed a general trend of significant increases in gene expression upon CRISPR-Cas9 targeting of these uncharacterized CREs in three of the five elements tested (Supplementary Fig. 2d and Supplementary Data 4). Collectively, our functional tests confirm that uncharacterized CREs contain true silencer elements.

**Candidate silencer element predictions.** Recent studies showed that well-trained support vector machine (SVM) models can predict CREs from a given set of nucleotide sequences[52–54]. Using a gapped k-mer SVM (gkmSVM)[55,56], we trained the classifier based on MPRA functional screening data to find candidate silencer elements from untested uncharacterized CREs in K562 and other cell and tissue types (Fig. 3a). We chose the top 2000 uncharacterized CREs sequences with the lowest MPRA activity as a positive set, and the bottom 2000 uncharacterized CREs with highest MPRA activity as a negative set for the gkmSVM model. We trained the gkmSVM model on 80% of the data, and used the remaining 20% of data for testing the model. We checked the performance of the model on test data by generating the receiver operating characteristic (ROC) curve by plotting true positive rate versus false positive rate and the precision recall curve (PRC). The model accurately predicted the positive uncharacterized CREs on the test set and the model performance in terms of area under the curve (AUC) for ROC curve is 0.81 and for PRC is 0.76 (Fig. 3b, c). We then used this model for predicting candidate silencer elements across all cell types from the list of uncharacterized CREs. We chose the threshold for the gkmSVM score where the model's accuracy is maximum in order to classify the positive uncharacterized CREs from the negative set. The percent of gkmSVM-positive predictions for candidate silencers range from 45% to 78% for human cell types, and it varies from 55% to 85% for mouse cell types (Supplementary Fig. 3a–c and Supplementary Data 5). We found a significant decrease in reporter gene

expression for two out of three tested candidate silencer elements predicted by the SVM model compared to two random controls in K562 cells tested via reporter assays (Fig. 2e; Supplementary Data 3).

Next, we determined what features drove the SVM predicted silencers. To do so, we examined whether positively predicted candidate silencers by the SVM model are significantly enriched with known repressor TFs motifs in comparison with negatively predicted ones. Motifs belonging to REST, GFI1B, NKX TFs, BAPX1, and CUX2 are significantly enriched consistently across all cell types. In addition, SOX TFs, ZNF TFs, ZBTB12, SMAD3, and TCF4 are also enriched across many cell types (Fig. 3d). Previous studies reported that many of these TFs display repressor activity[57–64]. The enriched motifs are distinct from all uncharacterized CREs (Fig. 1d).

In total, we predicted 1,706,989 candidate silencer elements in human genome spanning across 82 cell and tissue types from the Roadmap[27] and ENCODE consortia[28], and 965,198 elements in the mouse genome for 22 cell and tissue types from the ENCODE consortium[28] (Fig. 3e, g; Supplementary Fig. 3d and Supplementary Data 1). On average, ~33,600 elements and ~60,200 elements per cell type were identified in human and mouse genomes, respectively. We further filtered to identify cell-type-specific candidate silencer elements. In total, 1,230,490 cell-type specific candidate silencers in human cell types and 772,535 cell-type specific elements in mouse cell types are predicted (Fig. 3f, h; Supplementary Fig. 3e). The identified candidate silencer elements are largely present at intergenic, introns, and repeat elements of the genome for both human and mouse (Fig. 3i, j) and located relatively close to gene TSS (Supplementary Fig. 3f, g).

**Characterization of candidate silencer elements.** We performed motif enrichment analysis to identify potential transcription factors binding at candidate silencer elements from all cell types compared with matched random genomic background. As expected, motifs belonging to REST are highly enriched in candidate silencers of all 52 tested Roadmap human cell types (Fig. 4a, b). Motifs of TFAP2C, NF1, BATF, BACH2, FRA1, ATF3, FOSL2, ZFX, EBF1, NFE2, and RFX family of TFs are also enriched at candidate silencers of many cell types. Previous studies reported that these TFs display repressor activity[29–32,65,66]. We furthermore performed TF motif enrichments at these candidate silencer elements compared with active enhancers and DHS elements to check if candidate silencers are distinguished from other categories of CREs. We find that motifs belonging to REST, ZFX, PITX1, ZNF family and many other TFs such as CUX1, FOSL1, GFI1B, and GATA family of TFs are significantly enriched, indicating that these elements contain silencer elements (Supplementary Fig. 4a, b).

Next, we explored additional characteristics of the candidate silencer elements. DNA cytosine methylation is an important epigenetic modification affecting gene expression patterns during development and disease. DNA hypermethylation at promoters and CpG Islands often results in repression of gene expression and hypomethylation results in activation. Similarly, enhancer elements are either largely unmethylated or lowly methylated[67,68]. However, a number of transcriptional repressors are known to bind methylated DNA[69–71], suggesting some silencer elements would be methylated. We computed the average methylation levels for candidate silencer elements using whole genome bisulfite sequencing (WGBS) data from 17 cell types and found them to largely be in a hypermethylated state (Fig. 4c). We also compared the average methylation of active enhancers (DHS marked with H3K4me1 + H3K27ac) and all DHS elements for corresponding cell types and found significantly (P-value

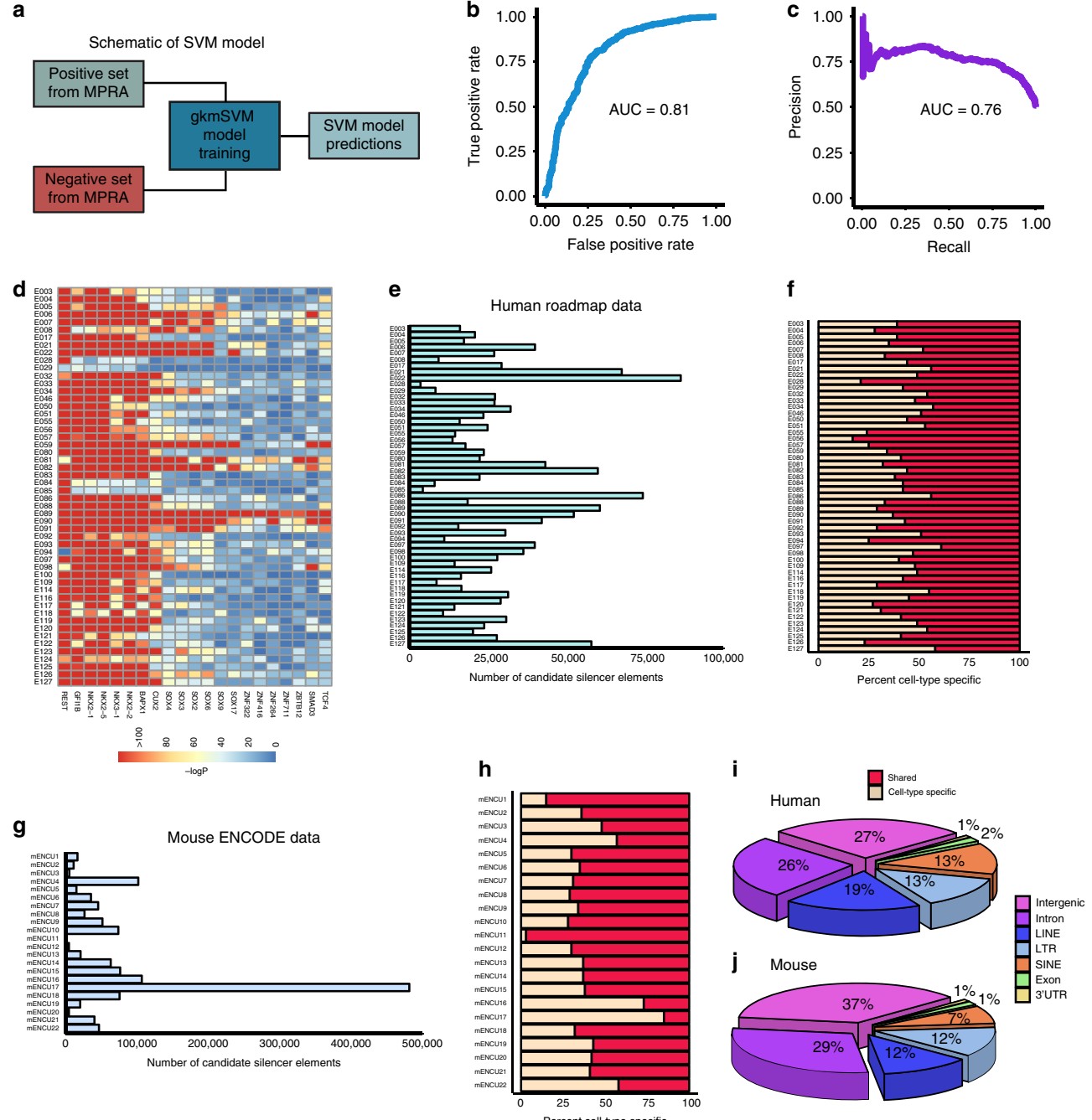

**Fig. 3 Overview of SVM model and candidate silencer element predictions. a** Schematic overview of the gkmSVM model development for candidate silencer predictions. The trained gkmSVM model performance on test data shown by receiver operating characteristic (ROC) curve (**b**) and precision recall curve (**c**). **d** Heatmap of enrichment (-log(*P*-value)) of TF motifs at SVM positively predicted uncharacterized CREs (candidate silencers) relative to SVM negatively predicted uncharacterized CREs across 52 cell types and tissues from Roadmap. Barplots presenting the count of predicted candidate silencer elements across different cell types and tissues in human from Roadmap (**e**) and mouse genomes from ENCODE (**g**). Distribution of cell-type-specific candidate silencer elements in human (**f**) and mouse genomes (**h**). Pie chart showing the genomic distributions of candidate silencer elements in human (**i**) and mouse (**j**).

< 2.2e-16) lower methylation for enhancers and DHS than at candidate silencers. Candidate silencers are on average 71.5% methylated compared with 43% methylation in active enhancers, and 46% methylation in DHS elements across the 17 cell types (Fig. 4c).

Enhancers and promoters can be annotated in the genome based on distinct chromatin signatures[72]. Such a chromatin signature for silencers would facilitate their genomic localization.

To investigate such, we made use of already available chromHMM annotations for 52 human cell types from the Roadmap consortium. Unfortunately, the majority of the candidate silencer elements belong to a largely uncharacterized annotation category or lack of known histone modification signal (quiescent: Quies) (Fig. 4d). This uncharacterized category varies from 27% to 81% across cell types. The next three categories are weak transcription (TxWk) and weak polycomb repressor

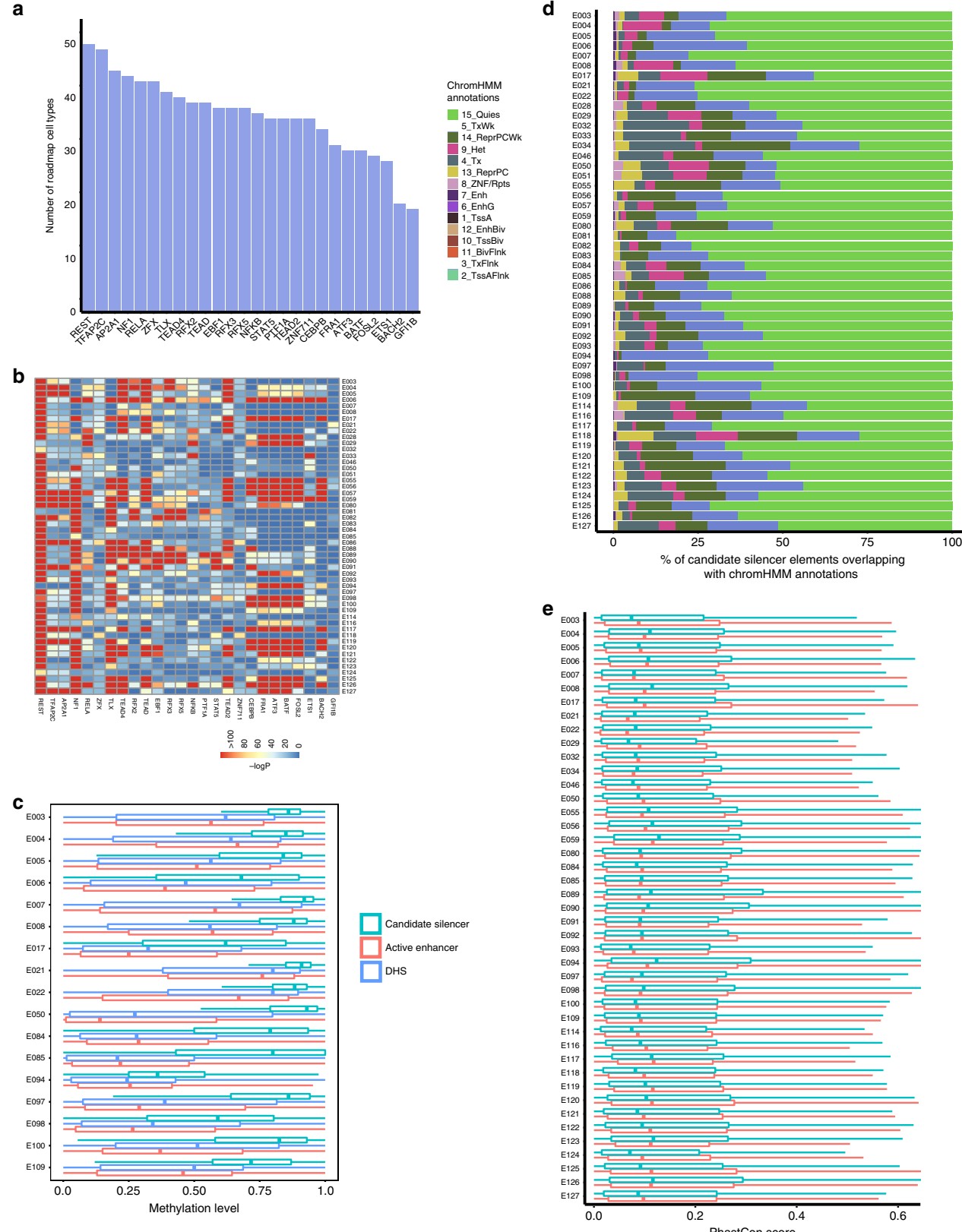

**Fig. 4 Characteristics of candidate silencer elements. a** Bar plot presenting the count of cell types enriched with TF motifs at candidate silencer elements across 52 cell types and tissues from Roadmap. **b** Heatmap of enrichment (-log($P$-value)) of TF motifs at candidate silencer elements across 52 cell types relative to random genomic background. **c** Box plot showing the distribution of methylation levels at candidate silencer elements, DHS and active enhancers across 17 cell types. **d** Bar plot showing the distribution of chromHMM annotations of candidate silencer elements across 52 human cell types. **e** Box plot of PhastCon conservation scores (seven ways) for candidate silencers and active enhancers across 52 cell types. In the box plots, bounds of the box spans from 25 to 75% percentile, center line represents median, and whiskers visualize 5 and 95% of the data points.

complex (ReprPCWk) followed by heterochromatin (Het). While limited, all three categories are consistent features of transcriptional repression and some repressor TFs. In addition to chromHMM annotations, we also looked at the silencer overlaps with repressive regions marked with H3K27me3 and H3K9me3 histone modifications. We found that on average ~2.7% and ~3.3% of silencer elements are present at H3K27me3 and H3K9me3 regions, respectively, and ~5.9% of elements present at one of these regions. In this analysis, we also checked whether candidate silencers are distinguished from other random accessible elements by comparing their chromHMM annotations. The random accessible elements are enriched with enhancers (Enh and EnhG), active TSS (TssA), flanking active TSS (TssAFlnk), and to some extent with uncharacterized category (Quies) (Supplementary Fig. 4d–i). This analysis also supports that the predicted candidate silencers are distinct from other CREs.

Lastly, we examined conservation scores to check whether candidate silencer elements are conserved in vertebrate evolution. We used the seven-way phastCons score comparing the human genome to six other vertebrate genomes. The computed phastCons scores suggest that candidate silencer elements are moderately conserved across species in orders of magnitude similar to enhancer elements[73] (Fig. 4e). The average phastCon score for candidate silencers is 0.183, whereas for active enhancers it is 0.177 across 52 cell types.

**Candidate silencer elements interact with inactive genes**. Previous promoter-capture HiC (p-CHiC) studies reported that promoters of transcriptionally inactive or lowly expressed genes are interacting with uncharacterized regions of genome, which suggests that they may act as silencers[51,74]. We investigated whether our predicted candidate silencer elements are interacting with inactive genes. For this analysis, we obtained five p-CHiC data sets, three in human (GM12878, CD34+ and H9 cells), and two in mouse (mESCs and mouse fetal liver cells (FLCs)) cell types (Fig. 5; Supplementary Fig. 5)[51,74]. We overlapped respective candidate silencer elements with non-baited fragments (or promoter-interacting fragments) to find their interacting genes. We exclusively focused on interactions that were separated by a distance of at least 10 kb between gene promoter and promoter-interacting fragments. This criterion removes the high frequency of close-proximity ligation events in the HiC data. In total, we find 12,321 candidate silencers from GM12878 are interacting with 17,250 genes in GM12878 cells and 5907 candidate silencers from CD34+ cells are interacting with 8599 genes in CD34+ cells. We find 4631 and 1621 inactive genes (with RPKM = 0) are interacting with 8329 and 1993 candidate silencers in GM12878 and CD34+ cells, respectively (Fig. 5a, d; Supplementary Fig. 5a, d). An additional set of 5918 and 3015 lowly expressed genes (with RPKM between >0 and 2) are interacting with 9412 and 3647 candidate silencer elements in GM12878 and CD34+ cells (Fig. 5a, d; Supplementary Fig. 5a, d). These results indicate that candidate silencers can interact with more than one gene and also multiple candidate silencers can interact with a single gene. Similar trends are observed in H9 cells as well (Supplementary Fig. 5g). Furthermore, the overall expression of all genes interacting with candidate silencers showed a significantly lower expression than genes interacting with active enhancers in both GM12878 and CD34+ cells (Fig. 5b, e). Next, we looked at the chromatin states of transcription start sites (TSS) of genes interacting with candidate silencers categorized into different expression ranges (Fig. 5c–f; Supplementary Fig. 5h). The majority of inactive gene TSS are uncharacterized with no known chromatin mark (Quies: GM12878 − 50%; CD34 − 52%; H9 −

59%) and to some extent enriched with weak polycomb repressor complex (ReprPCWk: GM12878–22%; CD34–17%; H9–5%), weak transcription (TxWk: GM12878–10%; CD34–6%; H9–7.5%), followed by heterochromatin (Het: GM12878–3%; CD34–3%; H9–1.3%). These chromatin states might be expected for genes repressed through distal silencers. In contrast, highly expressed genes are enriched with active TSS (TssA: GM12878–43%; CD34–30%; H9–55%) and transcription (Tx: GM12878–21%; CD34– 30%; H9–30%) annotation categories. The fraction of uncharacterized and active TSS are in opposing trends with increasing gene expression (Fig. 5c, f; Supplementary Fig. 5h).

We repeated the p-CHiC analysis with the mouse data sets and found more profound results (Fig. 5g–j). Briefly, a total of 6707 candidate silencers from mESCs are interacting with 5445 genes in mESCs and 2107 candidate silencers from FLC are interacting with 2619 genes in FLCs. We find 4220 and 1968 genes with RPKM in the range of 0–1 are interacting with 5674 and 1770 candidate silencers in mESCs and FLCs, respectively (Fig. 5g, i; Supplementary Fig. 5i, j). Overall, 81% and 79% of all silencer interacting genes are within the expression range of 0–2 RPKM in mESCs and FLCs, respectively (Fig. 5g, i; Supplementary Fig. 5i, j). Silencer interacting genes showed overall lower expression than enhancer interacting genes in both cell types (Fig. 5h, j). Altogether, this analysis provides strong evidence for silencer activity for the candidate silencer elements.

**Disease-associated variants are enriched at silencer elements**. Several earlier studies showed that disease-associated single-nucleotide polymorphisms (SNPs) from genome-wide association studies (GWAS) are prevalent at noncoding regions of the genome, especially at CREs, and often in enhancers of cells thought to be associated with the disease[27,75–77]. We next investigated whether candidate silencer elements were also enriched for disease-associated variants, as this would alter how the functional impact of the variant is interpreted for distal CREs. First, we determined how many SNPs are present at candidate silencer elements. For this, we downloaded all disease SNPs from the NHGRI-EBI GWAS catalog. We included both lead SNPs and SNPs in linkage disequilibrium (LD) ($r^2 \geq 0.8$) for overlap with candidate silencer elements. We found that 57,961 SNPs belonging to 2214 disease traits are present at silencer elements across all cell types. Next, we examined significant enrichment of specific disease SNPs at silencers for each cell type. This analysis resulted in ~20% of these diseases (451 out of 2214) being significantly enriched (adjusted $P$-value < 0.01, hypergeometric test) at silencers for one or more cell types (Fig. 6a, b; Supplementary Fig. 6a, b). Next, we asked whether these disease SNPs are enriched in relevant disease cell types. As a test case, we examined whether enrichment of autoimmune diseases are specific to blood cell types (Fig. 6c; Supplementary Fig. 6c, d). Indeed, we found systemic sclerosis, rheumatoid arthritis, asthma and hay fever, type 1 diabetes, and amyloid A serum levels to be specifically enriched at candidate silencer elements of blood cell types. Overall, our results illustrate that candidate silencer elements are also enriched with disease-associated SNPs similar to other cis-regulatory elements.

**Silencer elements can act as enhancers in other cell types**. We asked whether identified candidate silencer elements in one cell type act exclusively as silencers or can also enhance gene expression, i.e., act as active enhancers in other cell types. To verify this, we intersected the candidate silencer elements from each cell type with active enhancers (overlapping DHS marked with H3K4me1 + H3K27ac) of the remaining cell types.

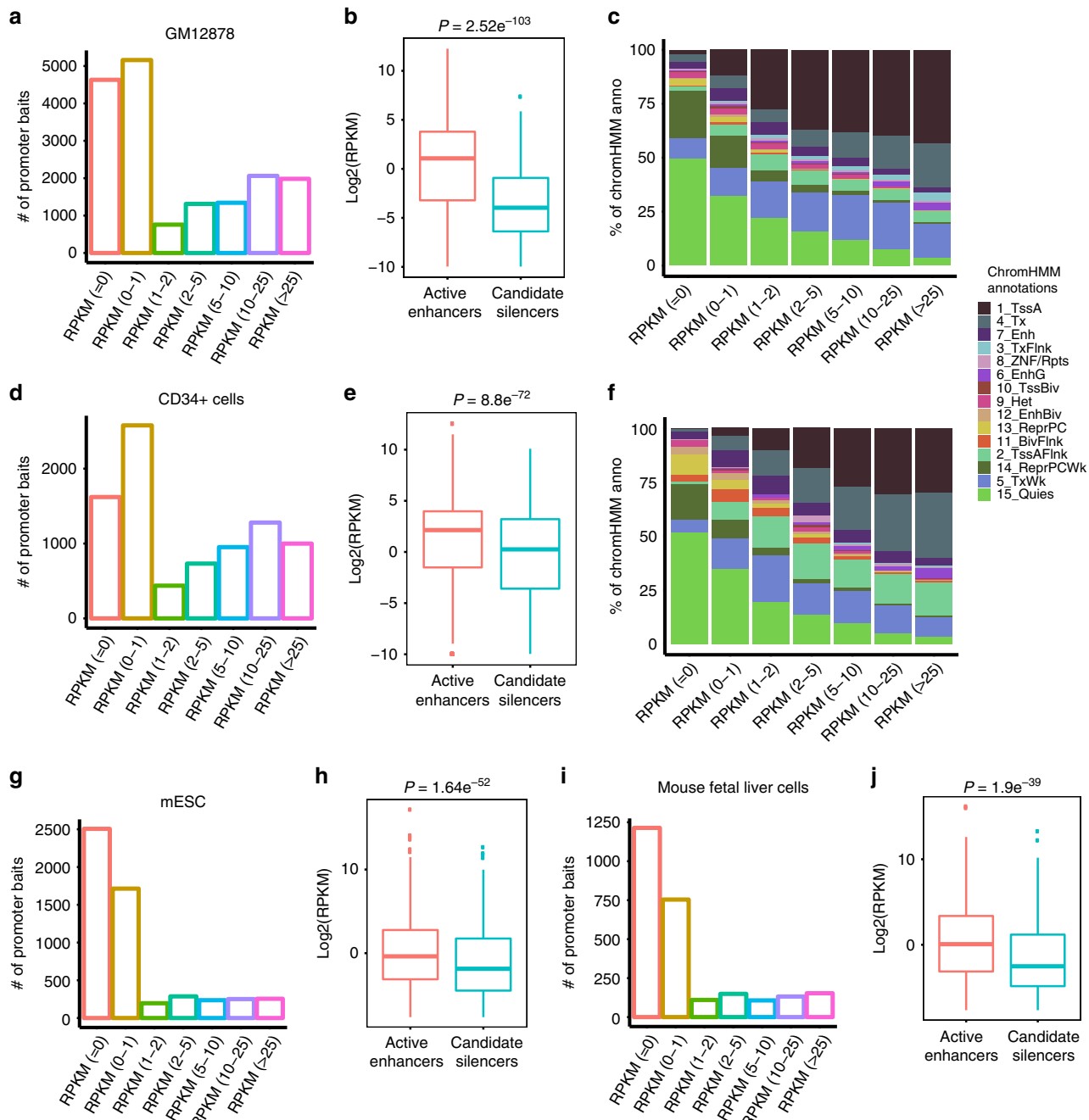

**Fig. 5 3D Genome interactions of candidate silencer elements. a** Bar plot presenting the counts of genes expressed at different expression levels (RPKM) interacting with candidate silencer elements in GM12878 cells. **b** Box plot comparing the expression levels (RPKM) of genes interacting with candidate silencer elements and active enhancers in GM12878 cells. **c** Bar plot showing the distribution of chromHMM annotations for TSS of genes interacting with candidate silencer elements in GM12878 cells. **d** Bar plot presenting the counts of genes expressed at different expression levels (RPKM) interacting with candidate silencer elements in CD34+ cells. **e** Box plot comparing the expression levels (RPKM) of genes interacting with candidate silencer elements and active enhancers in CD34+ cells. **f** Bar plot showing the distribution of chromHMM annotations for TSS of genes interacting with candidate silencer elements in CD34+ cells. **g** Bar plot presenting the counts of genes expressed at different expression levels (RPKM) interacting with candidate silencer elements in mouse embryonic cells (mESCs). **h** Box plot comparing the expression levels (RPKM) of genes interacting with candidate silencer elements and active enhancers in mESCs. **i** Bar plot presenting the counts of genes expressed at different expression levels (RPKM) interacting with candidate silencer elements in mouse fetal liver cells (FLCs). **j** Box plot comparing the expression levels (RPKM) of genes interacting with candidate silencer elements and active enhancers in FLCs. P-values computed by one-tailed Wilcoxon rank-sum test. In the box plots, bounds of the box spans from 25 to 75% percentile, center line represents median, and whiskers visualize 5 and 95% of the data points.

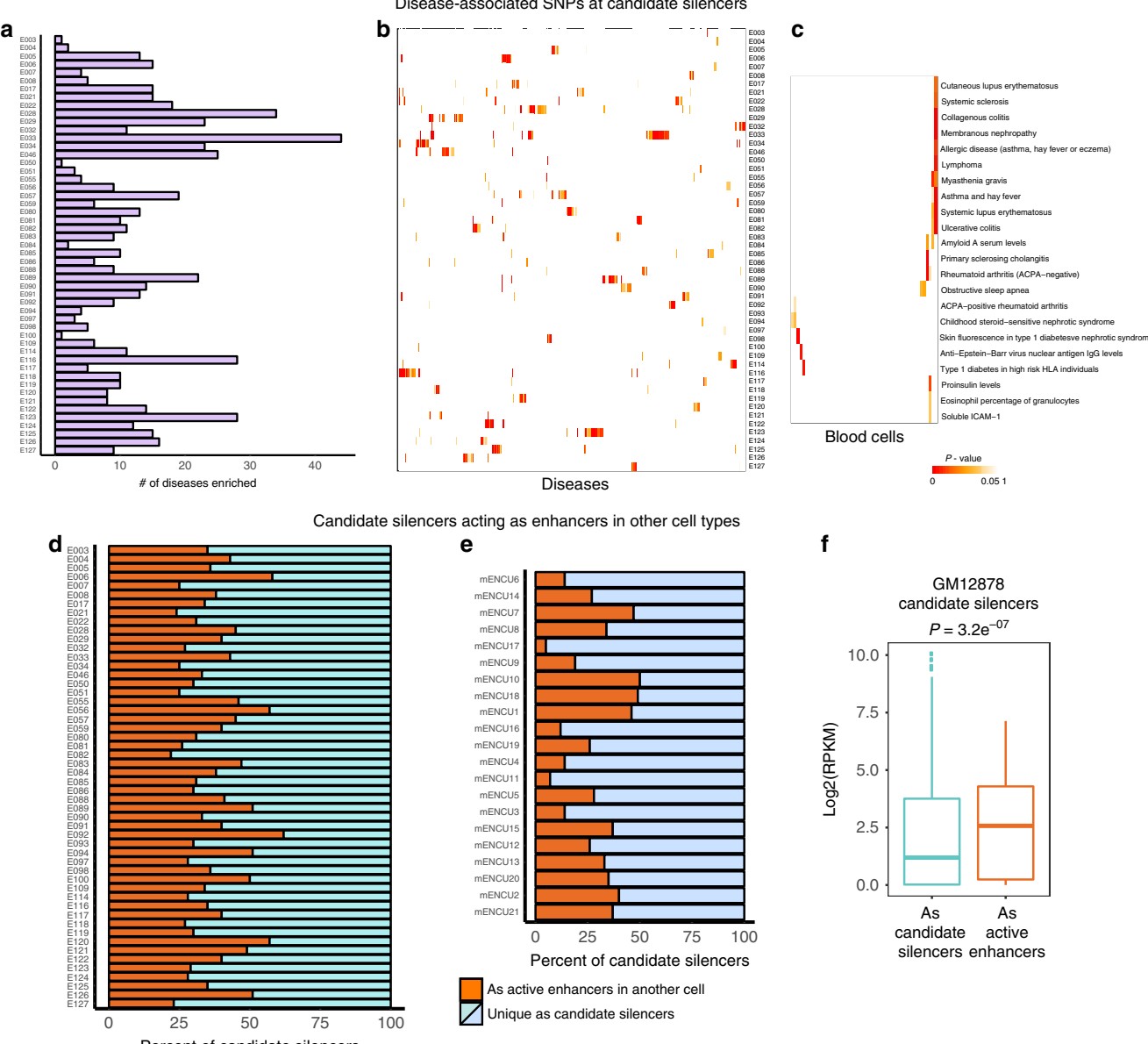

**Fig. 6 Candidate silencer elements are enriched for GWAS SNPs and can act as enhancers in other cell types. a** Bar plot presenting the count of GWAS SNPs belonging to various disease traits enriched (adj.*P*-value < 0.01) at candidate silencer elements of 52 cell types and tissues from Roadmap. **b** Heatmap of enrichment (-log10(*P*-value)) of 451 disease traits SNPs across 52 cell types. Columns represent disease traits and rows are cell types. **c** Heatmap of SNP enrichment (-log10(*P*-value)) for autoimmune disease traits across blood cell types. Bar plot showing the fraction of candidate silencer elements identified in one cell type acting as enhancers in other cell types for human Roadmap data (**d**) and mouse ENCODE data (**e**). **f** An example of GM12878 silencers acting as enhancers in CD34$^+$ cells. Box plot comparing the expression distribution of interacting genes. *P*-values computed by one-tailed Wilcoxon rank-sum test. In the box plots, bounds of the box spans from 25 to 75% percentile, center line represents median, and whiskers visualize 5 and 95% of the data points.

Interestingly, we found that 22–62% of candidate silencer elements from one cell type can act as enhancers in at least one other cell type (Fig. 6d). For the mouse candidate silencer elements, it varies from 5% to 50% (Fig. 6e). This accounts for 25.3% and 15.5% of all human and mouse candidate silencers, respectively. We hypothesize that silencers need a limited set of TFs compared to active enhancers for regulating the transcriptional output. In order to verify this feature, we made use of high-resolution digital genomic footprint (DGF) data, which indicates TF binding at single-nucleotide resolution at regulatory elements. First, we compared the number of DGFs per silencer and active enhancer elements for the same cell type across cell types and found on average lower DGFs per silencer than active enhancer

(Supplementary Fig. 6e). Next, we compared the DGFs per element at candidate silencers in one cell type and switching to enhancers in other cell types and found on average lower DGFs per element when acting as a silencer than acting as an enhancer in other cell types (Supplementary Fig. 6f). Next, we wanted to verify this feature of silencers acting as enhancers by looking at their interacting gene expression distribution. As a test case, we identified candidate silencer elements in GM12878 cells that overlap with (or switch to) active enhancer elements of CD34$^+$ cells. We then found enhancer interacting genes in CD34$^+$ cells from p-CHiC data as was done for silencer interacting genes in GM12878 cells and compared the target gene expression levels in their respective cell types. The relative expression of silencer

interacting genes in GM12878 cells is indeed significantly less compared with the expression of interacting genes in CD34$^+$ cells when silencers are acting as enhancers (Fig. 6f). Promoter elements frequently harbor multiple, yet distinct, regulatory sequences, including the core promoter, enhancer elements, and silencer elements. Our analysis of candidate distal silencers elements indicates that a subset can likely function as or are part of the same *cis*-regulatory element as enhancers.

## Discussion

The precise control of gene expression, either activation or repression, is essential for changes in cell fate and cellular response to external cues. That control is ultimately mediated by CREs in the genome and their cognate binding factors. Though silencers have shown to be analogs to enhancers in that they can be distal to the genes they regulate and often function in a position- or orientation-independent manner[4], silencers lack a unique chromatin signature to aid their genome-wide identification. Here, we developed a simple computational (SSA) approach to first identify genome-wide uncharacterized CREs containing candidate silencer elements. Then, functional screening via MPRAs was performed on a select subset of 7500 uncharacterized CREs to identify the true silencer elements. MPRA assays confirmed silencer activity for 41.5% of tested uncharacterized CREs. This validation rate is within the range or above what was shown for enhancer functional validation studies[26,78]. However, it is susceptible to false negatives due to caveats of in vitro reporter assays. It is also feasible that weak silencers were not detected given that the origin of replication can act as a promoter and also need to be repressed[79]. As further evidence of silencer activity, we tested a handful of silencer elements via traditional reporter assays, mitigating any effect of being in the 3′ UTR, and also demonstrated via CRISPR-Cas9 experiments that silencer deletions trend toward increased target gene expression.

Based on our MPRA data, we trained an SVM model to predict candidate silencer elements from untested uncharacterized CREs in K562 and other cell and tissue types. We find that candidate silencer elements are enriched with many known repressor TF motifs. P-CHiC interactions showed that inactive and lowly expressed genes are interacting with candidate silencer elements. These results strongly support the annotation of silencer elements in the human and mouse genomes. We acknowledge that this may still be an overestimate as the majority of these candidate silencer elements are model predicted, and large-scale functional validation screens are needed to obtain an accurate count of silencers. In addition, some cell-, tissue-, or species-specific silencers may be missed. Furthermore, these predictions also rely on the annotation of DHS in each genome, which are subject to some degree of both false positive and false negative calls.

The vast majority of disease-associated SNPs are known to occur outside of coding regions[80]. Similar to enhancers, we found disease-associated SNPs are also enriched at candidate silencer elements of relevant disease cell types or tissues[27]. Mutations within silencer elements may provide one means for genes to escape repression in a disease-specific manner and have implications for other diseases such as various cancers.

Overall, we find that the characteristics of the predicted candidate silencer elements are expected of silencer elements and are in contrast to that of active enhancers. The catalogs of candidate silencer elements presented here across many cell types and tissues for human and mouse may serve as a resource that complements the ENCODE[28] and Roadmap consortia[27] catalogs for other CREs. This approach should also prove applicable to future data sets from other cell types and across several species.

## Methods

**Processed data sets used**. We obtained uniformly processed consolidated epigenome data of ChIP-seq and DNase-seq from the Roadmap consortium for 52 human cell types. We used narrow peaks called by MACS2 program for DNase-seq and broad peaks called by MACS2 program for H3K4me1, H3K4me3, and H3K27ac histone modifications. We directly obtained the peaks files from NIH Epigenomics Roadmap project[27] (http://egg2.wustl.edu/roadmap/web_portal/processed_data.html). Similarly, we obtained the data for mouse (mm10) cell types and another 30 human cell types from the ENCODE consortium (https://www.encodeproject.org/matrix/?type=Experiment)[28]. We used the peak files generated by the uniform ENCODE Processing Pipeline. We merged the peaks files for each cell type if data are available from different sources. The CTCF TFBS are obtained from CTCFBSDB 2.0 (http://insulatordb.uthsc.edu/) for both human and mouse[81]. The gene TSS coordinates were obtained from GENCODE annotations for human (GRCh37.p13/hg19) and mouse (GRCm38.p5/mm10)[82].

**Simple subtractive analysis approach**. In the SSA approach, we used DNase-seq peaks as open chromatin (DHS), H3K4me3 peaks, and TSS plus 2000 bp upstream and 500 bp downstream of TSS of all GENCODE genes as promoter elements, H3K4me1 peaks as enhancer elements, and CTCF TFBS as insulator elements. We start with DHS peaks and removed DHS overlapping with any of enhancer elements or promoter elements or insulator elements in a cell-type specific manner, and the remaining non-overlapping DHS peaks are termed as uncharacterized CREs. We included CTCF TFBS from all cell types during subtractive analysis as these TFBS are largely shared across cell types. We used the BEDTools suite for genomic subtractive analysis[83]. The large variations in the number of uncharacterized CREs across cell types is most likely due to the quality of available chromatin data.

**Cell culture**. All experiments are performed in K562 cells. Cells obtained from the ATCC (CCL-243) are grown at 37 °C and 5% $CO_2$ in RPMI-1640 (Gibco) medium with 10% fetal bovine serum (Gibco) and 1% penicillin–streptomycin.

**Massively parallel reporter assays (MPRAs)**. To multiplex testing for silencer activity of uncharacterized CREs, we leveraged synthetic oligonucleotide array synthesis and adopted the STARR-seq (self-transcribing active regulatory region sequencing) method as described in ref. [41]. To select control-random regions, first we downloaded histone marks and DHS data in terms of processed peaks for K562 cells from the Roadmap Consortium. Next, we randomly selected genomic regions as control elements not overlapping with any of those annotated regions or any of Refseq annotations (promoters, genes, 3′UTRs, 5′UTRs, exons, introns, transcripts, CDS) as to avoid CREs as controls, as these were selected separately. Control enhancer and control silencer elements were obtained from a previous study[26]. An oligonucleotide library was synthesized containing 200 nt of genomic regions with 15 nt of flanking sequence matching the Illumina primer sequence (Agilent, Inc). The obtained library was PCR amplified using primer "Starr-seq_homology_forward: TAGAGCATGCACCGGAATGATACGGGCGACCACCGAGATCTACACT CTTTCCCTACACGACGCTCTTCCGATCT" and "Starr-seq_homology_reverse: GGCCGAATTCGTCGACAAGCAGAAGACGGCATACGAGAT[6-base-barcode] GTGACTGGAGTTCAGACGTGTGCTCTTCCGATC" to add Illumina primer sequence and 15 bp of sequence matching the STARR-seq backbone (Addgene: 71509) for In-Fusion cloning (Clontech). The library was amplified with KAPA Hifi 2×(Kapa biosystem) with the following thermal condition (98 °C for 2 min, amplification with ten cycles of 98 °C for 20 s, 65 °C for 15 s, and 72 °C for 30 s; final extension at 72 °C for 2 min). The resulting product was purified using Ampure-XP beads at 1.8× beads: reaction ratio. The STARR-seq screening vector was digested for 6 h with SalI-HF and AgeI-HF and the linearized backbone was run on a gel and purified with a gel purification kit (Qiagen). In total, 200 ng of backbone and 50 ng of pooled insert were cloned in four 10 µl Infusion-HD reactions incubating at 50 °C for 15 min (Clontech). Resulting products were then combined and purified using Ampure-XP beads with a 1× volume of beads and eluted in 8 µl of purified water and electroporated into NEB® 10-beta electrocompetent cells at a ratio of 2 µl of reaction to 20 µl of competent cells for a total of four electroporations using a Bio-Rad GenePulserR II electroporator, with the following electroporation conditions: 2.0 kV, 200 Ω, 25 µF. Transformations were recovered for 1 h in SOC medium while shaking (220 rpm, 37 °C), and then grown for 16 h in 500 mL of Luria Broth while shaking (220 rpm, 37 °C). The STARR-seq input library was then purified using the Qiagen Plasmid Maxi prep kit. K562 cells were electroporated with a Neon transfection kit and device (Invitrogen) with the following transfection parameters: pulse voltage: 1450; pulse width: 10; pulse number: 3. Five replicate transfections were performed. Cells were grown in antibiotic-free media for 24 h. Cell pellets were rinsed once with PBS, and then lysed in 2 ml of RLT buffer (Qiagen) with 2-mercaptoethanol. The total RNA was prepared using the QIAGEN RNeasy Plus kit. Poly-A RNA was isolated from 50 µg of the total RNA using the µMACS mRNA isolation kit. RNA was treated with turboDNase (4U) for 30 min at 37 °C. DNase-treated poly-A RNA was purified using the RNeasy kit. mRNA and RT primer "RT-primer: CAAACTCATCAATGTATCTTATCATG" were incubated at 65 °C for 5 min, and cDNA was synthesized using Superscript III by incubating for 1.5 h at 55 °C then inactivated at 80 °C for 15 min. Following synthesis, cDNA

was treated with RNaseA (Sigma) at 37 °C for 30 min. cDNA was purified using Ampure beads in a 1.5:1 bead:cDNA ratio and then amplified and indexed for sequencing using a two-stage PCR as described previously[41]. The cDNA sample from each replicate was used as an input into the first round gene-specific PCR using primers "targeted_library_F: GGGGCCAGCTGTTGGGGGTG*T*C*C*A*C" and "targeted_R: CTTATCATGTCTGCTCGA*A*G*C", and input sample from each replicate was amplified using primers "targeted_input_F: GGGGCCAGCTGT TGGGGTG*A*G*T*A*C" and "targeted_R: CTTATCATGTCTGCTCGA*A*G*C" and KAPA Hi-fidelity polymerase. PCR conditions were: 98 °C for 2 min, amplification with 12 cycles of 98 °C for 20 s, 65 °C for 20 s, and 72 °C for 60 s; final extension at 72 °C for 2 min. * = Phosphorothioated DNA bases.

Samples were then purified using the Zymo PCR purification kit and eluted in 15 μl of nuclease-free water. The resulting products were used as templates for the second round of PCR, which used a standard Illumina TruSeq indexing primer on the p5 end of the library and custom indexing primers to barcode the samples for multiplexing prior to sequencing (98 °C for 2 min, amplification with six cycles of 98 °C for 15 s, 65 °C for 30 s, and 72 °C for 30 s; final extension at 72 °C for 2 min. Final sequencing libraries were purified with Ampure-XP beads (Beckman Coulter) at a 1.8× SPRI: PCR ratio. All libraries were sequenced on Illumina NextSeq 500 performing 1 × 75 cycles.

**MPRA data normalization and analysis**. We generated data in five biological replicate measurements. Sequencing raw reads from RNA and plasmid libraries are checked for adapter sequences and low quality reads (q-score < 20) using FASTQC (https://www.bioinformatics.babraham.ac.uk/projects/fastqc/) and trimmed using TrimGalore package (https://www.bioinformatics.babraham.ac.uk/projects/trim_galore/). Trimmed reads were mapped to human genome (hg19) using the Bowtie2 aligner[84]. The mapped reads were quantified against all tested sequences using "featureCounts" function from the Rsubread package[85]. The biological replicates were checked for similarity. We retained tested sequences for downstream analysis only if it contain reads at minimum three biological replicates at both RNA and plasmid libraries. The RNA and plasmid read counts were normalized by respective library size, and the normalized silencer element activity (fold change) is defined as ratio of RNA to plasmid counts averaged over biological replicates divided by mean ratio of RNA to plasmid counts of random control regions for each tested sequence. One-tailed *t* test was used to find significant differences in tested silencer activity over control-random regions activity. The computed *P*-values are adjusted with Benjamini–Hochberg method to control for false discovery rate. Then, the potential silencers were identified as sequences with fold change less than one and adjusted *P*-value < 0.05. We compared the relative activity of tested uncharacterized CREs with that of control enhancers and control silencers and statistical significance were assessed using two-tailed *t* test.

**Reporter assays**. Reporter libraries: The STARR-seq luciferase validation vector_SCP1_empty (Addgene: 99299)[79] was digested for 6 h with BslI, and a linearized vector was purified by gel purification kit (Qiagen) and treated with Calf Intestine Alkaline Phosphatase (CIP). In total, 100 ng of backbone and 20 ng of gel-purified PCR product of candidate regions having 15 bp flanking for vector insert were cloned using Infusion-HD reactions by incubating at 50 °C for 15 min (Clontech). Resulting products were electroporated into NEB® 10-beta electrocompetent cells at a ratio of 2 μl of reaction to 20 μl of competent cells using a Bio-Rad GenePulserR II electroporator, with the following electroporation conditions: 2.0 kV, 200 Ω, 25 μF. Transformations were recovered for 1 h in SOC medium while shaking (220 rpm, 37 °C) and then grown under ampicillin selection. To avoid false positives, colony PCR was performed for the selected colonies using site-specific primers sets. The PCR-confirmed colonies were then grown for 16 h in 50 mL of Luria Broth with ampicillin selection while shaking (220 rpm, 37 °C). The input libraries were then purified using the Qiagen Plasmid Midiprep kit.

qPCR method: In all, 1 × 10^6 K562 cells were electroporated with 7.2 μg of the STARR-seq luciferase validation (firefly luciferase) reporter plasmid and 800 ng of *Renilla* luciferase control plasmid (pGL4.74 ([hRluc/TK]). A total of six silencer element STARR-seq plasmids, two random control plasmids, and an empty vector plasmid were tested in three independent transfections. qPCR to quantify firefly luciferase transcripts normalized to *Renilla* luciferase transcripts was performed 24 h after transfection. Cells were lysed using QIAshredder columns, and the total RNA was extracted using the RNeasy miniprep kit, with beta-mercaptoethanol supplemented RLT buffer. In all, 1 μg of the total RNA was treated with recombinant DNaseI for 30 min at 37 °C followed by the removal of rDNaseI using a DNase inactivation reagent. The DNaseI-treated RNA was reverse transcribed using Invitrogen's Superscript IV Vilo kit (25° for 10 min, 50 °C for 10 min, 85 °C for 5 min), followed by qPCR on 2 μl of diluted (1:5) cDNA using Go Tag SYBR Green qPCR Master Mix in a total volume of 10 μl with 0.5 μM gene-specific qPCR primers (95 °C, 2 min; 95 °C, 3 s; 60 °C, 30 s; 40 cycles total, see the Table provided below for primers)[79].

Table for qPCR primers:
FF_fwd: GTGGTGTGCAGCGAGAATAG
FF_rev: CGCTCGTTGTAGATGTCGTTAG
RL_fwd: CAACTACAACGCCTACCTTCG
RL_rev: CGGTGTTAGGGAACTTCTTAGCTC

qPCR analysis for reporter assay: Firefly luciferase Ct values for each candidate region were normalized to Renilla firefly Ct values using delta-Ct method[86]. Delta-delta-Ct values were calculated between silencer elements and random controls were displayed as average (2^{-ΔΔct}).

**CRISPR-Cas9 mediated characterization of candidates**. The CRISPR-Cas9 system was used to edit potential silencer regions in K562 cells. Briefly, three gRNAs spanning each silencer region and one non-targeting control gRNA (targeting lambda-phage DNA) were designed using CCTop[87], (https://crispr.cos.uni-heidelberg.de/index.html). CCTOP predicted target sgRNA were selected according to their best hit/least off-target parameters (pattern: N20NGG, core length = 12, max. core mismatches = 2, max. total mismatches = 4). Each gRNA template was ordered in the form of Gblocks from IDT and cloned into the lenti Guide puro vector (Addgene: #73795). gRNA plasmids along with SpCas9-HF1 (Addgene: #72247) were transfected using the Neon electroporator. Twenty-four hours post transfection, puromycin selection was applied for 48 h. Cells were further transduced using adenovirus vector expressing Cas9 (kindly provided by Andre Lieber's laboratory) at 2000 MOI. Ninety-six hours post transfection, RNA isolation was performed with TRIzol™ according to the manufacturer's protocol. cDNA synthesis were carried out using SuperScript™ IV VILO™ Master Mix with the ezDNase kit. To determine the effect on the expression of candidate target genes, primers were selected from primer bank (https://pga.mgh.harvard.edu/primerbank/). See Supplementary Data 4 for a list of primers and guideRNA sequences. qRT-PCR was performed on three biological replicates consisting of three technical replicates each. QuantStudio™ 3 System with KAPA SYBR® FAST qPCR master mix was used to calculate Ct value. Ct values were normalized using two internal references 18S rRNA & GAPDH RNA. Delta-Ct values were compared between experimental condition and control to calculate the differences in expression status. Statistical significance was assessed using one-tailed Student's *t* test. Our current experimental design is limited in that we are examining the effects of deletions in a highly heterogeneous (polyclonal background) population and used only a single non-targeting control gRNA targeting lambda-phage DNA.

**SVM model development and silencer element predictions**. We used the R package "gkmSVM" for SVM model training and candidate silencer element predictions from untested uncharacterized CREs in K562 cells and other cell types[56]. We chose the top 2000 candidate silencer sequences with lowest activity as a positive set and bottom 2000 silencers with highest activity as a negative set for SVM model training with default settings from the package. We used 80% of data for training, and the remaining 20% data for testing the model. We evaluated the performance of the model on test data by generating the ROC curve by plotting true positive rate versus false positive rate and the precision recall curves (PRC). We then used this model for predicting candidate silencers across all cell types from the list of uncharacterized CREs. We chose the threshold for the gkmSVM score where the model's accuracy is maximum in order to classify the positive uncharacterized CREs, i.e., candidate silencer elements from negative ones.

**Genomic annotation of candidate silencer elements**. The genomic annotations of predicted candidate silencers in human and mouse are performed using the HOMER suite[88] (*annotatePeaks.pl*).

**Candidate silencer elements overlap with repressor TFBS**. The well-known repressor TFBS such as REST, YY1, ZBTB33, SUZ12, and EZH2 for cell types GM12878, H1, K562, and HEPG2 are directly downloaded from ENCODE project as processed peak files. We computed the enrichment (number of overlaps) of repressor TFBS at candidate silencer elements using BedTools suite[83].

To compute the statistical significance for enrichment of candidate silencer elements at repressor TFBS relative to random DHS and random enhancers, we performed 10,000 random permutation tests and computed *P*-values. Briefly, we randomly selected the same number of DHS elements or enhancers as silencer elements and overlapped with repressor TFBS and compared these overlaps with silencer overlaps. Then, *P*-value is computed as follows:

$$P = \frac{\Sigma n + 1}{N + 1}. \tag{1}$$

Where, $\Sigma n$ is the number of permutations where expected overlaps (random DHS or enhancer overlaps with repressor TFBS) are greater than observed overlaps (silencer elements overlaps with repressor TFBS) and $N$ (=10,000) is the total number of permutations. The data in figures are represented as mean ± s.d.

**Motif enrichment analysis**. Motif analysis to find enriched TF motifs at candidate silencer elements relative to matched random genomic background were performed with HOMER suite (*findMotifsGenome.pl* -size given). In addition, we also identified enriched motifs at silencer elements relative to all DHS elements and enhancers within the same cell type as backgrounds. The enrichment of *P*-values are computed using binomial distribution. Enriched motifs are defined at adjusted *P*-value < 0.001. Heatmap visualizations are created using R package "pheatmap" and "ggplot2".

**Computing average methylation values**. The methylation value for each cytosine from WGBS data for 17 cell types were obtained from NIH Epigenomics Roadmap project as bigwig files. The bigwig files were converted to bedgraph files via UCSC file conversion tool (*bigWigToBedGraph*), and we computed the average methylation levels across candidate silencer elements, DHS peaks and active enhancers.

**chromHMM annotation data sets**. The chromHMM annotation categories for all 52 human cell types are obtained from NIH Epigenomics Roadmap project. We used the core 15-state model for chromatin state learning for predicted candidate silencer elements and for TSS of genes interacting with candidate silencer elements. In addition, we also computed the overlaps of randomly selected DHS elements at chromHMM annotations. We repeated this process over 1000 random permutations and took the mean values over 1000 permutations.

**Computing average phastCon conservation scores**. We computed average phastCons scores for candidate silencer elements and active enhancers using the R/ Bioconductor package "phastCons7way.ucsc.hg38" with "scores" function.

**Promoter-capture HiC data sets used**. We obtained promoter-capture HiC data for GM12878 and CD34[+] cells in human from ref. [51] and H9 cells from ref. [89] and mouse embryonic cells (mESCs) and mouse fetal liver cells (FLCs) from ref. [74]. The processed significant promoter—other end-fragment interactions were directly obtained from the authors' supplementary data. The corresponding promoter gene expression data for GM12878 and CD34[+] cells are obtained from NIH Epigenomics Roadmap project, and for mESC and FLC are obtained from the ENCODE project. To find the target genes interacting with candidate silencer elements, we overlapped silencer elements with other end fragments. Then, we looked at the expression distribution of those target genes in the respective cell type. Similarly, we overlapped active enhancers with other end fragments to find enhancer interacting genes. We defined active enhancers as DHS overlapping both H3K4me1 and H3K27ac (DHS + H3K4me1 + H3K27ac).

**Disease SNPs analysis**. Disease-associated SNPs were obtained from NHGRI GWAS catalog (https://www.ebi.ac.uk/gwas/docs/file-downloads) dated March 2019, and SNP coordinates are converted from GRCh38 to hg19 using UCSC liftOver tool. We included both lead SNPs and SNPs in LD block ($r^2 > = 0.8$) for overlap with silencer elements. The SNPs in LD (proxy SNPs) were obtained from SNAP web server (http://archive.broadinstitute.org/mpg/snap/ldsearch.php)[90]. The enrichment of disease SNPs to candidate silencer elements for each cell type are obtained by comparing the relative enrichment of particular disease SNPs against the rest of disease SNPs using a hypergeometric distribution.

**Statistical tests and visualizations**. All the statistical tests are performed in the R environment (https://www.r-project.org/). Graphs and visualizations are prepared using "ggplot2" and "gplots" R packages.

**Reporting summary**. Further information on research design is available in the Nature Research Reporting Summary linked to this article.

## Data availability
The sequencing data generated by this study have been deposited into NCBI Gene Expression Omnibus (GEO) under accession GSE142207. All processed data, including catalogs of candidate silencers, are made available through the Open Science Framework [https://osf.io/hzc3p/]. All other relevant data supporting the key findings of this study are available within the article and its Supplementary Information files or from the corresponding authors upon reasonable request.

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

## Acknowledgements

R.D.H. is supported by funds from the NIH/NIAMS (R01AR065952), NIH/NIDDK (R01DK103667).

## Author contributions

N.D.J. and R.D.H. conceived and planned the study, and wrote the paper. N.D.J. performed all the analyses. A.J. performed the MPRAs and reporter assays. A.M. performed CRISPR-Cas9 experiments. All authors reviewed the final paper.

## Competing interests

The authors declare no competing interests.
