## [Peer Review File · Nature Communications]

REVIEWERS' COMMENTS:

Reviewer #5 (Remarks to the Author):

The authors have mostly addressed the concerns I raised in a previous round of review, and I recommend this for publication.

The updated Fig 2b is much more convincing.

I would strongly recommend including the exact sequences/coordinates of all of the tested control and silencer regions in the supplement, if it is not already included.

I remain concerned about the CRISPR experiments, regarding examining the effects of deletions in a highly heterogeneous / polyclonal background and using only a single control gRNA. I would recommend that the authors explain the experimental setup more fully in the text / figure legend / methods, and mention the limitations of this approach. The authors could also consider moving these data (Fig 2f) to a supplementary figure.

Doni Jayavelu et al.

Please see our response below to Reviewer 5's comments.

Reviewer #5 (Remarks to the Author):

The authors have mostly addressed the concerns I raised in a previous round of review, and I recommend this for publication.

The updated Fig 2b is much more convincing.

I would strongly recommend including the exact sequences/coordinates of all of the tested control and silencer regions in the supplement, if it is not already included.

I remain concerned about the CRISPR experiments, regarding examining the effects of deletions in a highly heterogeneous / polyclonal background and using only a single control gRNA. I would recommend that the authors explain the experimental setup more fully in the text / figure legend / methods, and mention the limitations of this approach. The authors could also consider moving these data (Fig 2f) to a supplementary figure.

Response: We thank the reviewer for positive feedback. We have provided the exact sequences with coordinates for all tested elements (control enhancers, control silencers, control random regions and uncharacterized CREs) with their raw read counts and measured activity levels in Supplementary Data 2. We moved Fig. 2f to the supplement (now Supplementary Fig. 2d) as suggested by the reviewer and provided experimental design in complete details and also mentioned the limitations in the text and figure legend.